# Ten Thousand-Fold Higher than Acceptable Bacterial Loads Detected in Kenyan Hospital Environments: Targeted Approaches to Reduce Contamination Levels

**DOI:** 10.3390/ijerph18136810

**Published:** 2021-06-25

**Authors:** Erick Odoyo, Daniel Matano, Martin Georges, Fredrick Tiria, Samuel Wahome, Cecilia Kyany’a, Lillian Musila

**Affiliations:** 1United States Army Medical Research Directorate-Africa, Nairobi P.O. Box 606-00621, Kenya; Erick.Odoyo@usamru-k.org (E.O.); Martin.Georges@usamru-k.org (M.G.); Fredrick.Tiria@usamru-k.org (F.T.); cecilia.katunge@usamru-k.org (C.K.); 2Kenya Medical Research Institute, Nairobi P.O. Box 54848-00200, Kenya; dmatano@kemri.org; 3Independent Researcher, Nairobi P.O. Box 64-20300, Kenya; minorsamwah@gmail.com

**Keywords:** infection prevention and control, bacterial loads, high-touch surfaces

## Abstract

Microbial monitoring of hospital surfaces can help identify target areas for improved infection prevention and control (IPCs). This study aimed to determine the levels and variations in the bacterial contamination of high-touch surfaces in five Kenyan hospitals and identify the contributing modifiable risk factors. A total of 559 high-touch surfaces in four departments identified as high risk of hospital-acquired infections were sampled and examined for bacterial levels of contamination using standard bacteriological culture methods. Bacteria were detected in 536/559 (95.9%) surfaces. The median bacterial load on all sampled surfaces was 6.0 × 10^4^ CFU/cm^2^ (interquartile range (IQR); 8.0 × 10^3^–1.0 × 10^6^). Only 55/559 (9.8%) of the sampled surfaces had acceptable bacterial loads, <5 CFU/cm². Cleaning practices, such as daily washing of patient sheets, incident rate ratio (IRR) = 0.10 [95% CI: 0.04–0.24], providing hand wash stations, IRR = 0.25 [95% CI: 0.02–0.30], having running water, IRR = 0.19 [95% CI: 0.08–0.47] and soap for handwashing IRR = 0.21 [95% CI: 0.12–0.39] each significantly lowered bacterial loads. Transporting dirty linen in a designated container, IRR = 72.11 [95% CI: 20.22–257.14], increased bacterial loads. The study hospitals can best reduce the bacterial loads by improving waste-handling protocols, cleaning high-touch surfaces five times a day and providing soap at the handwash stations.

## 1. Introduction

Healthcare-associated infections (HAIs) lead to prolonged hospital stays, increased mortality [1,2], and an increase in the overall cost of healthcare [3]. The rates of HAIs are disproportionately higher in African countries, estimated to range from 2.5% to 14.8% [4]. In Kenya, the rate of HAIs is estimated to be 4.4 per 100 patient admissions [5]. HAIs are frequently caused by pathogens that contaminate and persist in the hospital environment [6]. They can be transmitted to the patient through direct contact with the contaminated environment or contamination of healthcare workers’ gloves and hands [7]. HAIs arising from surface contamination by bacterial pathogens, including *Clostridioides difficile*, *Acinetobacter baumannii*, oxacillin-resistant *Staphylococcus aureus*, carbapenem-resistant Enterobacterales, and vancomycin-resistant enterococci, have been reported [8]. Hospital surfaces, such as sinks, water dispensers, patients’ beds, and linens contaminated with these pathogenic bacteria, have been implicated in HAIs and hospital outbreaks [9,10,11]. 

The implementation of IPC practices could help lower the risk of HAIs. For instance, effective hospital environment cleaning can reduce surface bio-contamination and the subsequent risk of HAIs [12,13]. However, suboptimal cleaning of hospital environments has been reported in many hospitals [14,15], and surfaces remain contaminated with microbial pathogens even after cleaning and disinfection. This residual contamination is partly because many hospitals only conduct visual inspections of the hospital environments to evaluate the cleaning procedures, instead of biomonitoring [14,16,17]. An aerobic colony count of 5 CFU/cm² on high-touch surfaces in the hospital is considered to be the upper limit for acceptable hospital surface hygiene [8]. Colony counts >5 CFU/cm² indicate that the bacterial contamination is high enough to pose a threat of HAIs and highlights the need for improved infection control practices. While bio-surveillance was chiefly conducted during outbreaks to aid in targeted cleaning, hospitals with weak infection prevention systems could benefit from routine biomonitoring to expose and address inadequacies in their practices. The safe handling and disposal of biomedical waste, which potentially contains pathogens that can infect patients and healthcare providers, can also help to lower HAIs. However, deficiencies in the proper handling and management of biomedical waste, including unsafe injection equipment, have been reported in many developing countries [18], reflecting a significant risk for HAIs.

Kenya introduced IPC guidelines for health care services [19] in 2010, aiming to implement these practices within hospitals and subsequently reduce HAIs. However, several challenges remain in implementing these IPC strategies, including limited resources evidenced by shortages of protective equipment, lack of running water, and poor adherence to the recommendations reflected in poor hand hygiene practices among clinicians [20]. In many hospitals in Kenya, the magnitude of the problem, however, has not been measured. The microbial monitoring of specific hospital surfaces or equipment that is often touched or handled by patients and clinicians, regarded as high-touch surfaces, can help identify target areas for improved IPCs that could help lower HAIs. For instance, by monitoring bacterial contamination levels on high-touch surfaces, the effectiveness of cleaning and disinfection practices in the hospital can be determined. This study was conducted to determine levels and variations in bacterial contamination of high-touch surfaces of five Kenyan hospitals and to determine their contributing modifiable risk factors. The study has quantified bacterial loads and identified target areas for improved infection control, which could significantly reduce the HAI risks.

## 2. Materials and Methods

### 2.1. Research Frame

This study was conducted in five hospitals in Kenya in which four departments in each hospital, with high HAIs, were selected for sampling. High-touch surfaces were sampled and evaluated for their bacterial levels. Additionally, IPC practices in each department were recorded. Statistical analysis was conducted to determine the distribution of bacterial levels and the influence of IPCs on the observed bacterial levels.

### 2.2. Study Design 

A descriptive laboratory-based study was conducted in five hospitals in Kenya: two level five hospitals (A and C), one level six hospital (B), and two level four hospitals (D and E). Hospital B is a national referral and teaching hospital offering specialized medical services with a 450-bed capacity. In contrast, hospitals A and C, with 168 and 270-bed capacities, respectively, provide a range of medical services and are referral hospitals for sub-county hospitals within their counties. Hospitals D and E, with 158 and 54-bed capacities, respectively, are primary care facilities. In each hospital, departments with high HAI levels were identified, and a member of the hospital infection prevention committee completed a questionnaire to determine the hospital’s IPC practices. The questionnaire was composed of questions regarding the availability of personal protective equipment, running water, soap and hand wash stations, the segregation and cleaning of hospital garments, the frequency of cleaning or decontaminating high-touch surfaces and floors, detergents or decontaminants used, and biosafety practices. Four hospital departments were selected for sampling in each study hospital: three wards and the outpatient departments. Five outpatient departments (hospitals A, B, C, D and E), five paediatric (hospitals A, B, C, D and E), three maternity wards (hospitals C, D and E), three newborn units (hospitals A, B, and D), two general wards (hospitals D and E), two male surgical wards (hospitals A and B) and one female surgical ward (hospital A) were sampled.

### 2.3. Environmental Sampling

Sampling was carried out twice in each of the hospitals, between February and September 2018. In each selected department, the specific areas that patients and clinical staff frequently touched, termed “high-touch areas”, were sampled. These high-touch areas included items and surfaces close to the patient, such as patient beds, bed rails, baby cots, intravenous pole steering handles, newborn incubators, intravenous tubing, tray tabletops, baby weighing scale, bedside tabletops, room light switch plates, and room inner doorknobs. Other sampled areas were bathroom areas (bathroom sinks, toilet handrails, and flush), equipment (computer keyboards and mouse, clinician cell phones, blood pressure cuffs, stethoscopes, and thermometers), and clinician gowns (Appendix A). A sterile square frame measuring 10 cm^2^ was used to define the swabbed area to ensure uniform sampling for areas. For smaller objects or surfaces, the surface area was approximated, and the whole area was swabbed. Samples from clinician uniforms were collected by swabbing the abdominal region, sleeve cuffs, and coat pockets of the uniform. Swabs in neutralizing buffer (N.B.) (Puritan ESK sampling kit, Guilford, ME, USA) were used to sample 559 high-touch surfaces and equipment. The swabs were rotated with firm pressure over the target areas and then repeated at perpendicular angles for maximum recovery. One swab was used for each surface. The swabs were then shipped at 4 °C to the testing lab at the Kenya Medical Research Institute (KEMRI), Nairobi, and received for processing within 36 h.

### 2.4. Swab Processing 

The NB solution containing the swab was vortexed to release the bacteria from the swab. Two 1:10 serial dilutions were made from the N.B. suspension in sterile 0.01 M phosphate-buffered saline. A total of 100 µL of the original swab suspension and 100 µL of the two dilutions were then inoculated on 5% sheep blood agar (HiMedia Laboratories, Mumbai, India) using a sterile spreader and incubated at 37 °C for 18 h. Colonies were counted from the plate dilution with the most countable numbers, and colony-forming units/mL (CFU/mL) in the original sample were calculated. *Staphylococcus aureus* 25,923 and *Escherichia coli* 25,922 were used to perform quality control of the 5% sheep blood agar.

### 2.5. Statistical Analysis

All statistical analyses were carried out using STATA (StataCorp. 2013. College Station, TX, USA). Descriptive statistics were expressed as percentage, median and interquartile range (IQR) due to the non-normal distribution of the data. The Kruskal–Wallis H test for categorical data was used to determine if there were statistically significant differences in bacterial loads between the groups of independent variables. Univariate analysis was performed using negative binomial regression to assess the influence of the IPC practices on bacterial loads. The relative risks between different IPC practices were expressed as the incidence rate ratio (IRR). For multivariate analysis, the least absolute shrinkage and selection operator (LASSO) with cross-validation was used for model selection by estimating the model coefficients used to select covariates included in the model. Split-sampling and goodness of fit were used to ascertain that the features found can be generalized outside the training sample. LASSO was used to select the IPC practices that provided accurate information on the response variable. The chosen practices were included in a negative binomial regression analysis to estimate their influence and significance on bacterial loads. A *p*-value of less than 0.05 was considered statistically significant. Confidence intervals (C.I.s) of 95% were calculated for both univariate and multivariate analysis.

## 3. Results

### 3.1. Recovery of Bacteria from the High-Touch Surfaces

Four departments were sampled at two separate times in each of the five hospitals, yielding 559 samples. Bacterial growth was detected in 536/559 (95.9%) surfaces; bacterial counts were obtained in 513/559 (91.8%) surfaces, while 23/559 (4.1%) surfaces had swarming growth and could not be enumerated. A total of 23/559 (4.1%) surfaces sampled had no bacterial growth. The median bacterial load in all sampled surfaces was 6.0 × 10^4^ CFU/cm^2^ [IQR; 8.0 × 10^3^–1.0 × 10^6^]. Overall, only 55/559 (9.8%) of the surfaces sampled had acceptable bacterial loads of <5 CFU/cm².

### 3.2. Bacterial Loads across the Study Hospitals

Across the five sampled hospitals, the median bacterial load was highest in hospital C, 1.7 × 10^5^ CFU/cm^2^ [IQR; 2.25 × 10^4^–1.18 × 10^6^], followed by hospital B, 1 × 10^5^ CFU/cm^2^ [IQR; 8.8 × 10^4^–1 × 10^6^], with the lowest median bacterial load found in hospital A, 2.3 × 10^4^ [IQR; 5.0 × 10^3^–2.45 × 10^5^] (Figure 1). Hospitals D and E had median bacterial loads of 5.6 × 10^4^ [IQR; 1.1 × 10^4^–1.9 × 10^6^] and 2.7 × 10^4^ [IQR; 1.0 × 10^3^–3.0 × 10^5^], respectively. A significant difference in terms of median bacterial loads was reported across the hospitals (*p* < 0.001). In hospitals C and B, the bacterial loads were disbursed throughout the range of values observed, while in hospitals A, D, and E, the bacterial loads were mostly clustered at levels > 2.0 × 10^5^ and <5.0 × 10^4^.

### 3.3. Bacterial Loads across the Sampled Hospitals Departments

Maternity departments had the highest median bacterial loads 3.8 × 10^5^ CFU/cm^2^ [IQR; 5.35 × 10^4^–7.65 × 10^6^], followed by the general wards 1.13 × 10^5^ CFU/cm^2^ [IQR; 1.1 × 10^4^–1.0 × 10^6^]. Female surgical wards and the outpatient departments had the lowest bacterial loads, 2.0 × 10^4^ CFU/cm^2^ [IQR; 1.0 × 10^4^–1.6 × 10^5^] and 2.0 × 10^4^ [IQR; 2.0 × 10^3^–27 × 10^5^], respectively (Figure 2). Most of the bacterial counts in the maternity department were greater than 2.0 × 10^5^, while most bacterial loads in outpatient and female surgical departments were lower than 5.0 × 10^4^. The bacterial loads were dispersed throughout the range of values observed within the general wards, male surgical wards, newborn, and paediatric departments.

### 3.4. IPC Practices in the Hospitals 

All the sampled high-touch surfaces were routinely cleaned using bleach (sodium hypochlorite) with or without soap. The median bacterial loads for surfaces routinely cleaned with bleach in combination with soap 311/559 (55.6%) were significantly lower than those cleaned with bleach alone, 248/559 (44.4%) [*p* < 0.002]. Additionally, the median bacterial loads for sampled high-touch surfaces in departments where soap was available at the handwash stations, 377/559 (67.4%), and coats were available for clinician use, 404/559 (72.3%), were significantly lower, *p* = 0.01 and, *p* = 0.02, respectively (Table 1). Significantly lower median bacterial loads were also found in departments where mops were stored wet, 84/559 (15.0%), *p* = 0.02, and where gowns were washed within the facility, 169/559 (30.2%) [*p* = 0.01]. Most hospital floors and high-touch surfaces were cleaned twice a day, 341/559 (61.0%) and 270/559 (48.3%), respectively. Significant differences were observed in the median bacterial loads for the different frequencies of cleaning and/or decontamination of the high-touch surfaces (*p* = 0.004) and the floor (*p* = 0.004). 

### 3.5. Impact of Biosafety Practices on Bacterial Loads in the Study Hospitals

Significantly higher median bacterial loads were found in departments where dirty linen was transported in designated containers 524/559 (93.7%) [*p* = 0.03], and biosafety waste was removed at least daily from the area 496/559 (88.7%) [*p* = 0.03]. Additionally, the median bacterial loads were significantly higher for the sampled high-touch surfaces in departments where containers for needle disposal were available 524/559 (93.7%) [*p* = 0.03] and where biosafety waste was collected in designated bins 523/559 (93.6%) [*p* = 0.03]. Significantly higher median bacterial loads were also found for sampled high-touch surfaces in departments where the movement of people was limited 366/559 (65.5%) [*p* = 0.01] and where beds were more than one meter apart 92/559 (16.5%) [*p* = 0.04].

There was no significant difference in the median bacterial loads in departments based on ventilation, the presence or absence of isolation rooms, and the location of storage of sterile instruments (Table 2). A total of 27/160 (16.9%) of the reusable equipment were not cleaned, and 6/160 (3.7%) of the sampled sterile instruments were also not appropriately sterilized and packed. However, the differences in median bacterial loads between these groups of factors were not statistically significant.

### 3.6. Influence of Cleaning Practices on Bacterial Loads in the Study Hospitals

Univariate analysis was performed using negative binomial regression, adjusting for clustering by hospitals to determine the influence of the infection control practices on bacterial loads (Table 3). Cleaning practices, such as the daily washing of patient sheets or gowns, IRR = 0.10 [95% CI: 0.04–0.24], providing hand-wash stations, IRR = 0.25 [95% CI: 0.02–0.30], having running water, IRR = 0.19 [95% CI: 0.08–0.47] and soap, IRR = 0.21 [95% CI: 0.12–0.39] at the hand-wash stations each significantly lowered the incidence of bacterial loads (Table 3). Storing mops wet instead of dry also considerably reduced the incidence of bacterial loads IRR = 0.08 [95% CI: 0.02–0.30] (*p * = 0.001). There was no significant decrease in bacterial load in departments, where routine cleaning was performed either with bleach alone or with bleach combined with soap (*p * = 0.53).

Increasing the frequency of cleaning the floor from once a day to thrice a day, IRR = 0.04 [95% CI: 0.01–0.10] or four times a day, IRR = 0.21 [95% CI: 0.07–0.59] decreased the bacterial loads. The cleaning of high-touch surfaces twice a day, IRR = 0.21 [95% CI: 0.06–0.76], thrice a day IRR = 0.11 [95% CI: 0.04–0.32] or five times a day, IRR = 0.02 [95% CI: 0.01–0.04] significantly lowered the incidence of bacterial loads compared to once a day (Table 3).

### 3.7. Influence of Storage, Disposal, and Biosafety Practices on Bacterial Loads in the Study Hospitals

Transporting dirty linen in a designated container, IRR = 72.11 (95% CI: 20.22–257.14), and retaining personal protective garments within the facility, IRR = 5.84 (95% CI: 3.09–11.02), both increased the incidence of bacterial loads in the study hospitals (Table 3). Biosafety practices, such as collecting waste in designated bins IRR = 74.1 (95% CI: 23.70–231.70), removing the biosafety waste at least daily from the department IRR = 71.80 (95% CI: 20.75–248.45), and having an isolation room IRR = 6.37 (95% CI: 1.61–25.16), significantly increased the incidence of bacterial loads.

### 3.8. Multivariate Analysis of the Influence of Hospital IPC Practices and Environmental Conditions on Bacterial Loads in the Study Hospitals

Multivariate analysis was performed using negative binomial regression, adjusted for clustering by hospitals, and controlled for departments to determine which infection control practices would best predict bacterial loads. Cleaning the floor thrice a day, IRR = 0.40 [95% CI: 0.17–0.90], cleaning frequently touched surfaces five times a day IRR = 0.64 [95% CI: 0.46–0.89], and providing soap at the hand wash station, IRR = 0.66 [95% CI: 0.50–0.86], significantly decreased the bacterial loads (Table 4). However, the presence of a hand wash station within the department, IRR = 0.88 [95% CI: 0.64–1.22], availability of hand sanitizers, IRR = 0.96 [95% CI: 0.69–1.32], and retaining personal protective garments in the facility IRR = 0.63 [95% CI: 0.23–1.72], as well as the collection of biosafety waste in designated bins, IRR = 0.87 [95% CI: 0.46–1.67], and restriction of movement into the departments, IRR = 0.98 [95% CI: 0.69–1.37] reduced bacterial loads, but were not statistically significant (Table 4).

## 4. Discussion

The implementation of IPC practices to reduce bacterial environmental contamination and bacterial transmission could help lower the risk of HAIs. In most hospitals in Kenya, the magnitude of environmental contamination has not been measured. This study systematically sampled and measured the levels of bacterial contamination in high-touch areas of five Kenyan hospitals. Only 9.8% of the sampled high-touch surfaces had acceptable bacterial loads, <5 CFU/cm². The median bacterial load across the study hospitals was 10,000-fold higher than the acceptable level. Although Hospital C (a level 5 hospital) had the highest median load (a hundred thousand-fold higher than acceptable), the loads were high across all hospitals, irrespective of the level of the hospital. The female surgical and outpatient departments, which had the lowest median bacterial load across the sampled hospital departments, were still 10,000-fold higher than acceptable levels. These consistently unacceptable microbial loads were observed despite all the hospital departments carrying out routine disinfection and cleaning the high-touch surfaces using either bleach alone or bleach and soap. In a previous study, the pediatric department was identified as having high rates of HAIs in Kenya [5]. This study found loads which were 10,000-fold higher than is acceptable in paediatric departments, mirroring this increased risk of HAIs. In practice, the cleaning and disinfection of the hospital’s high-touch surfaces should remove and prevent the establishment of bacteria from the hospital surfaces and lower HAI risk [21]. Therefore, it is likely that, in these study hospitals, either the bacterial loads were quickly re-established after cleaning to their former levels or substandard cleaning methods were used. Another explanation for this is that the cleaning materials used did not adequately remove the bacteria from the hospital surfaces and may have inadvertently seeded these surfaces with more bacteria, as previously observed [22]. Bleach is also often available in concentrations that require further dilutions to appropriate working concentrations before use, which is likely not attained in the study hospitals. 

This study examined the roles that different cleaning practices might play in the hospital’s bacterial loads. Cleaning practices such as the daily washing of patient sheets and gowns, and provision of handwash stations, running water, and soap at the handwash stations significantly lowered the incidence of bacterial loads. Increased frequencies of floor-cleaning coincided with a decreased incidence of bacterial loads. Cleaning high-touch surfaces five times a day had the most impact on reducing bacterial loads in the study hospitals. Increased cleaning frequencies may improve patient safety [13] and are encouraged for the study hospitals, as recommended in the Kenya guidelines on IPC practices for health care services [19]. However, increased cleaning frequencies may be unattainable for some of the hospitals because of the accompanying costs. A total of 17.4% of the hospital departments reported a lack of running water, reflecting either limited resource availability, poor water infrastructure at the municipal or county level, or a lack of knowledge of and adherence to IPC practices. Interestingly, low levels of hand hygiene in Kenyan hospitals were observed even when water was available for handwashing [23], which means that a multifaceted approach including behavioural change may be required to ultimately reduce the risk of HAIs in the country. Alcohol-based hand sanitizers, an alternative low-cost solution, may also improve healthcare worker compliance with hand hygiene and address the water shortages [24,25]. All the five study hospitals sampled had outsourced cleaning services. Outsourced hospital cleaning is a way to reduce healthcare service costs [26]. However, given the critical role of routine targeted cleaning in preventing HAIs, lack of training on which surfaces and cleaners to use, frequency of routine cleaning, absence of biomonitoring of bacterial loads and key environmental pathogens, the monitoring and feedback to cleaners [8] could lead to substandard cleaning practices, inadvertently increasing HAIs. 

For instance, mopping solutions and mop heads are usually quickly contaminated during cleaning [27], so it is advised to launder the mops heads after use and allow them to dry before reuse to avoid recontamination [28]. In practice, using a contaminated cleaning material may increase bacterial count [22,29]. Mops were used to clean the floors in all the study hospitals, and, as an independent factor, storing mops wet correlated with a decrease in bacterial loads. A wet mop in some hospitals could be associated with increased frequencies of cleaning the floor, which is a consistently strong predictor of reduced bacterial contamination. However, in the regression analysis, storing wet mops was predictive of increased bacterial loads, reflecting that this poor cleaning practice, found in 15% of the sampled departments, contributes more to the spread of bacterial contamination. Educational programs for hospital cleaners could help improve hospital cleaning and or decontamination.

Bacterial loads were significantly reduced in-hospital departments where gloves and coats or gowns were available for clinician use. Essential personal protective equipment such as gloves and clinician coats or gowns are minimum requirements for IPC practices. Some of the study hospitals, however, faced a shortage of these essential supplies. In the Kenyan IPC guidelines [19], it is recommended that healthcare workers, hospital support staff, laboratory staff, and family members who provide care to patients should don personal protective equipment, including gloves and gowns. This recommendation remains to be achieved for all hospitals, and the actual implementation of the proposed guidelines could help reduce bacterial loads in the study hospitals.

Restricting or limiting the movement of people into the hospital departments lowered the incidence of bacterial loads. Wherever practical, it is important to reduce the levels of bacterial recontaminations of hospital surfaces through contact, the settlement of airborne bacteria bound on dust particles, or spills of bodily fluids. Restricting or limiting the movement of people into departments can reduce the dispersal of dust particles with airborne-bound bacteria that could re-contaminate the hospital’s high-touch surfaces. While most departments reported ‘good ventilation’, the ventilation consisted of large open windows, close to the ground, and vents that lacked any mechanical barriers for dust particles. This design feature may explain the increased incidence of bacterial loads observed in departments that reported ‘good ventilation.’ It is possible that, in departments where the movement of people was not restricted and lacked any physical barriers for dust particles in their ventilation, increased frequencies of cleaning alone were not enough to achieve acceptable high-touch surface bacteria loads, as recontamination quickly occurred. Congestion in many hospitals, and particularly specialty departments like paediatric units, may also prevent adequate terminal cleaning between patients.

Poor biomedical waste management may result in infections acquired from patients and healthcare workers [30]. It is estimated that between 18 and 64% of hospital facilities in 22 developing countries do not have proper waste disposal methods [18]. The enforcement of rules relating to biomedical waste in developing countries remains difficult. The Kenya IPC guidelines recommend that biomedical waste is removed from the hospital areas at least three times a day, or more frequently, as needed, and in a designated container [19]. Practices such as the availability of containers for needle disposal, biosafety waste collected in designated bins, and biosafety waste being removed from the area at least daily all recorded positive responses greater than 88%. This high percentage contrasts with those seen for cleaning practices such as the availability of handwash stations and soap, which both had lower positive responses of 77.8% and 67.4%, respectively. IPC training in Kenya may have emphasized practices such as proper medical waste management and injection safety [31], which could have led to the high positive responses in the recommended biosafety practices, such as those observed in this study.

Furthermore, high levels of compliance with the recommendation on waste segregation of syringes and needles were observed in Kenyan hospitals in a previous study [23]. Despite most hospital departments following the recommended biosafety practices, this study found that wherever biosafety waste was removed from the departments at least once daily, and in designated containers, the incidence of bacterial loads was significantly higher. A higher load was also found when dirty linen was transported in a designated container and containers for needle disposal were available. These designated waste containers could act as sources of bacterial recontamination when these containers are not adequately sanitized to remove any bound dust particles, soil, or organic matter when returned to the hospital departments. It may also be possible that the designated containers in the study hospitals were not the recommended leak-proof closed containers [32], which adequately contain biomedical waste. These findings highlight the need to conduct comprehensive training on IPCs, as enforcing one practice in the absence of others may not lead to a decreased risk of HAIs.

As of the time of writing this article, it is more than a decade since the launch of the Kenya IPC guidelines [19]. These guidelines have hand hygiene, personal protective equipment, cleaning and decontamination procedures, and safe handling of sharps as key factors. The implementation of these guidelines has faced several challenges. Due to the absence of a systematic monitoring and evaluation system, it may not be possible to pinpoint areas for improvement or the gains achieved as a consequence of the guidelines. Indeed, several deficiencies in IPCs still exist, as evidenced by this study outcome. For instance, poor water infrastructure at municipal or county levels still hinders the practice of handwashing. All the study hospitals only conducted visual inspections, instead of bio-surveillance, which are inadequate for highlighting target areas for improved IPCs. Biosurveillance is mainly impractical because these hospitals do not have functional microbiology laboratories, which are often not prioritized when allocating the limited resources. While such an intervention may require substantial resource allocation to implement, this study also recommends simple interventions, such as providing hand wash stations and soap, which could have an immediate impact, reducing the risk of HAIs in the hospitals. The high levels of adherence to biosafety practices, including the availability of containers for needle disposal, daily disposal of waste, and availability of designated containers for waste disposal, observed not only in this study but in another study in other Kenyan hospitals [23], highlight the gains achieved since the implementation of the Kenya IPC guidelines. This study provides a basis for updating the Kenya IPC guidelines to reduce HAIs.

### Limitations of the Study

The study had a few limitations. A uniform number of high-touch surfaces across the hospitals and departments were not sampled, given the differences in the departments, resources available, and services offered in each hospital, preventing more direct inter-hospital and inter-departmental comparison. Samples were collected at different times across the study hospitals, so other seasonal factors that could influence bacterial loads could have affected the outcomes. Not all IPC practices that could impact hospital bacterial loads were studied. Nonetheless, this study conducted one of the few systematic environmental bio-surveillance studies in Kenya to assess microbial loads and highlight target areas for improved hospital IPC to reduce bacterial loads. This study was a quantitative analysis of microbial loads that did not indicate bacterial pathogens which pose a real risk to patients. A qualitative study to determine the identity and antimicrobial susceptibility patterns of the contaminating bacterial pathogens was also conducted and will be published to identify the real HAI risk within these hospitals. 

## 5. Conclusions

A lack of compliance with the established IPC guidelines and recommendations was observed across all the study hospitals, explaining the very high microbial loads detected across the five hospitals. It is likely that a number of these study hospitals not only had limited resources but also lacked the knowledge to implement the recommended IPC practices, or did not consistently adhere to the IPC guidelines. Overall, the study shows that the study hospitals could best reduce bacterial loads by cleaning the floor thrice a day, cleaning high-touch surfaces five times a day, providing soap at hand wash stations, and providing gloves and gowns to hospital workers. Water, hygiene, and sanitation remain very important, and focussing on providing water and hand hygiene can significantly improve overall health and outcomes in hospitals. Alcohol-based hand sanitizers can be used as an alternative, low-cost solution to help achieve hand hygiene within hospitals, to overcome water or hand wash station shortages. Additionally, educational programs for housekeepers regarding effective cleaning practices could help lower the hospital bacterial loads. Routine biomonitoring can also expose inadequacies in cleaning practices and is encouraged in the study hospitals. The findings of this study are encouraging because they demonstrate that making small changes can significantly impact bacterial loads, which could reduce HAIs. 

## Figures and Tables

**Figure 1 ijerph-18-06810-f001:**
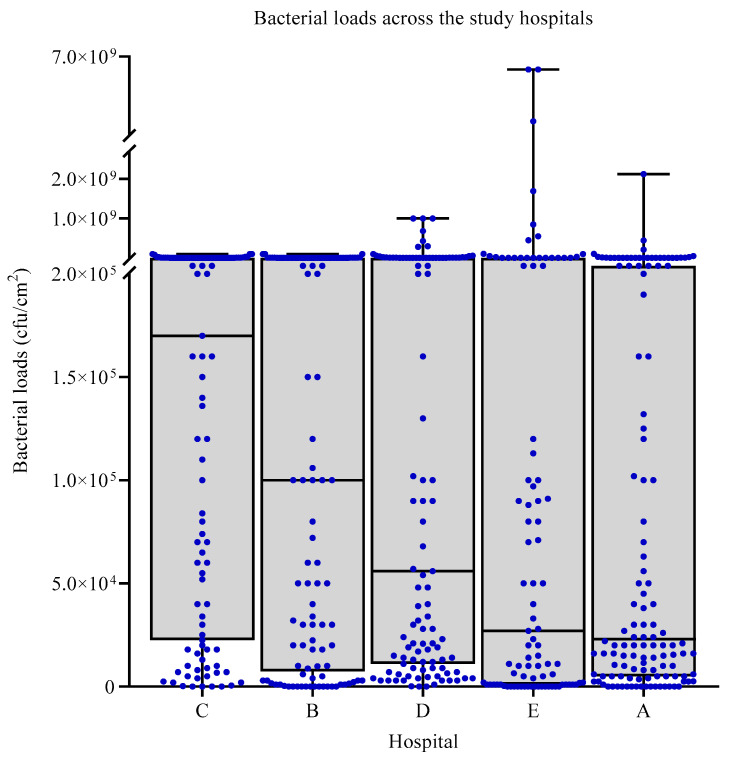
The bacterial load distribution among the study hospitals; in hospitals C and B, the bacterial loads are disbursed throughout the range, while in hospitals A, D, and E, the bacterial loads were mostly greater than 2.0 × 10^5^ and/or fewer than 5.0 × 10^4^.

**Figure 2 ijerph-18-06810-f002:**
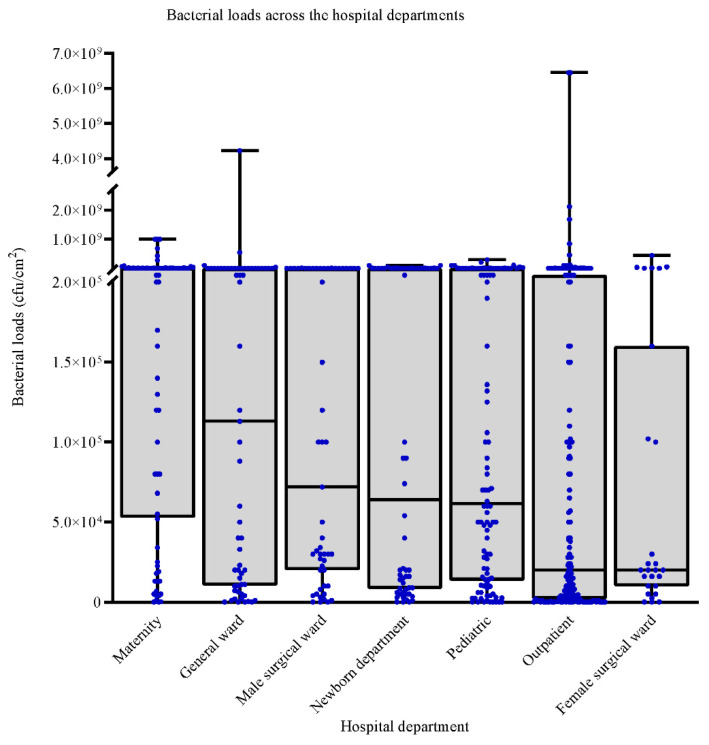
Median and IQR distributions of bacterial loads across the hospital departments; in the maternity department, the bacterial loads are mostly greater than 2.0 × 10^5^, while in the outpatient and female surgical departments, the bacterial loads are mostly lower than 5.0 × 10^4^. The bacterial loads are disbursed throughout the general wards, male surgical wards, newborns, and paediatric departments.

**Table 1 ijerph-18-06810-t001:** Characteristics of IPC practices and bacterial loads.

Hospital Infection Control Practice	Total No. of Items or Equipment Sampled (%) (*n* = 559)	Median Bacterial Loads (IQR)	*p*-Value
Detergents and/or disinfectants used to clean and/or decontaminate the hospital surfaces			
Bleach only	248 (44.4)	9.0 × 10^4^ (1.2 × 10^4^–1.0 × 10^6^)	**0.002**
Bleach and Soap	311 (55.6)	5.0 × 10^4^ (5.0 × 10^3^–5.0 × 10^5^)	
Frequency of cleaning floors daily			
Once	134 (24.0)	5.65 × 10^4^ (1.15 × 10^4^–1.3 × 10^6^)	**0.004**
Twice	341 (61.0)	5.0 × 10^4^ (5.0 × 10^3^–5.0 × 10^5^)	
Thrice	68 (12.2)	1.0 × 10^5^ (2.0 × 10^4^–1.0 × 10^6^)	
Four times	16 (2.8)	2.0 × 10^6^ (5.0 × 10^5^–1.0 × 10^7^)	
Frequency of cleaning high-touch surfaces daily			
Once	149 (26.7)	5.3 × 10^4^ (7.0 × 10^3^–1.0 × 10^6^)	**0.004**
Twice	270 (48.3)	7.0 × 10^4^ (9.8 × 10^3^–1.0 × 10^6^)	
Thrice	23 (4.1)	7.1 × 10^4^ (1.4 × 10^4^–8.0 × 10^5^)	
Four times	55 (9.8)	1.0 × 10^4^ (0–2.4 × 10^5^)	
Five times	42 (7.5)	1.5 × 10^5^ (3.0 × 10^4^–1.0 × 10^6^)	
Rarely	20 (3.6)	1.7 × 10^4^ (6.0 × 103–1.0 × 10^5^)	
Mops and cleaning cloths stored wet			
Yes	84 (15.0)	1.8 × 10^5^ (2.1 × 10^4^–2.0 × 10^6^)	**0.02**
No	475 (85.0)	5.0 × 10^4^ (6.6 × 10^3^–8.0 × 10^5^)	
Availability of running water			
Yes	462 (82.6)	6.4 × 10^4^ (9.0 × 10^3^–1.0 × 10^6^)	0.13
No	97 (17.4)	3.7 × 10^4^ (1.7 × 10^3^–7.3 × 10^5^)	
Availability of hand wash station in the department			
Yes	435 (77.8)	5.0 × 10^4^ (8.8 × 10^3^–1.0 × 10^6^)	0.94
No	124 (22.2)	9.0 × 10^4^ (5.0 × 10^3^–1.0 × 10^6^)	
Availability of soap at the handwash stations			
Yes	377 (67.4)	8.2 × 10^4^ (1.2 × 10^4^–1.0 × 10^6^)	**0.01**
No	182 (32.6)	2.8 × 10^4^ (3.0 × 10^3^–6.1 × 10^5^)	
Availability of gloves for clinician use			
Yes	504 (90.2)	7.0 × 10^4^ (1.0 × 10^4^–1.0 × 10^6^)	**<0.001**
No	55 (9.8)	1.0 × 10^4^ (0–2.4 × 10^5^)	
Availability of laboratory coats or gowns clinicians and support staff			
Yes	404 (72.3)	6.7 × 10^4^ (1.0 × 10^4^–1.0 × 10^6^)	**0.02**
No	155 (27.7)	5.0 × 10^4^ (1.6 × 10^3^–5.0 × 10^5^)	
Personal protective garments were retained in the facility			
Yes	94 (16.8)	5.0 × 10^4^ (1.0 × 10^3^–1.0 × 10^6^)	0.13
No	465 (83.2)	6.3 × 10^4^ (1.0 × 10^4^–1.0 × 10^6^)	
Clinician laboratory coats or gowns were washed within the facility			
Yes	169 (30.2)	2.1 × 10^4^ (3.5 × 10^3^–5,0 × 10^5^)	**0.01**
No	390 (69.8)	8.8 × 10^4^ (1.1 × 10^4^–1.0 × 10^6^)	
Patient sheets and gowns were washed daily or when soiled			
Yes	504 (90.2)	7.0 × 10^4^ (1.0 × 10^4^–1.0 × 10^6^)	**<0.001**
No	55 (9.8)	1.0 × 10^4^ (0–2.4 × 10^5^)	

**Bold** indicates significance at *p* ≤ 0.05.

**Table 2 ijerph-18-06810-t002:** Impact of biosafety practices on bacterial loads in the study hospitals.

Hospital Storage, Disposal, and Biosafety Practice	Total No. of Items or Equipment Sampled (%) (*n* = 559)	Median Bacterial Loads (IQR)	*p* Value
Dirty linen or gowns transported in a designated container			
Yes	524 (93.7)	6.3 × 10^4^ (9.0 × 10^3^–1.0 × 10^6^)	**0.03**
No	35 (6.3)	2.0 × 10^4^ (3.0 × 10^2^–2.2 × 10^5^)	
Availability of containers for needle disposal			
Yes	524 (93.7)	6.3 × 10^4^ (9.0 × 10^3^–1.0 × 10^6^)	**0.03**
No	35 (6.3)	2.0 × 10^4^ (3.0 × 10^2^–2.2 × 10^5^)	
Biosafety waste collected in designated bins			
Yes	523 (93.6)	6.15 × 10^4^ (9.0 × 10^3^–1.0 × 10^6^)	**0.03**
No	36 (6.4)	3.5 × 10^4^ (3.0 × 10^2^–2.2 × 10^5^)	
Biosafety waste removed at least daily from the area			
Yes	496 (88.7)	7.0 × 10^4^ (9.0 × 10^3^–1.0 × 10^6^)	**0.03**
No	63 (11.2)	2.0 × 10^4^ (3.5 × 10^3^–2.8 × 10^5^)	
Availability of quarantine or isolation rooms in the department			
Yes	302 (54.0)	8.0 × 10^5^ (6.75 × 10^3^–1.0 × 10^6^)	0.16
No	257 (46.0)	4.5 × 10^4^ (9.0 × 10^3^–4.8 × 10^5^)	
Beds greater than 1 m apart			
Yes	92 (16.5)	1.13 × 10^5^ (2.0 × 10^4^–1.18 × 10^6^)	**0.04**
No	467 (83.5)	5.0 × 10^4^ (6.0 × 10^3^–8.0 × 10^5^)	
The department was well ventilated			
Yes	337 (60.3)	7.0 × 10^4^ (9.0 × 10^3^–8.0 × 10^5^)	0.74
No	222 (39.7)	5.0 × 10^4^ (7.0 × 10^3^–1.2 × 10^6^)	
Movement of people limited or restricted into the departments			
Yes	366 (65.5)	8.0 × 10^4^ (1.0 × 10^4^–1.0 × 10^6^)	**0.01**
No	193 (34.5)	3.7 × 10^4^ (3.0 × 10^3^–5.0 × 10^5^)	

**Bold** indicates significance at *p* ≤ 0.05.

**Table 3 ijerph-18-06810-t003:** Univariate analysis of the influence of hospital IPC practices and environmental conditions on bacterial loads in the study hospitals.

Hospital Infection Control Practices and Environmental Conditions	IRR (95% C.I.)	Standard Error	*p*-Value
Detergents and/or disinfectants used to clean and/or decontaminate the hospital surfaces			
Bleach	0.67 (0.19–2.29)	0.42	0.53
Bleach/Soap	Reference		
Frequency of cleaning floors daily			
Once	Reference		
Twice	0.73 (0.21–2.50)	0.46	0.65
Thrice	0.04 (0.01–0.10)	0.02	**<0.001 ***
Four times	0.21 (0.07–0.59)	0.11	**0.004 ***
Frequency of cleaning high-touch surfaces daily			
Once	Reference		
Twice	0.21 (0.06–0.76)	0.14	**0.02 ***
Thrice	0.11 (0.04–0.32)	0.06	**<0.001 ***
Four times	4.18 (1.55–11.23)	2.1	**0.005 ^#^**
Five times	0.02 (0.01–0.04)	0.01	**<0.001 ***
Rarely	0.06 (0.02–0.16)	0.03	**<0.001 ***
Mops and cleaning cloths stored wet (Yes)	0.08 (0.02–0.30)	0.05	**<0.001 ***
Availability of running water (Yes)	0.19 (0.08–0.47)	0.08	**<0.001 ***
Availability of hand wash station in the department (Yes)	0.25 (0.07–0.94)	0.17	**0.04 ***
Availability of soap at the hand wash stations (Yes)	0.21 (0.12–0.39)	0.06	**<0.001 ***
Availability of gloves for clinician use (Yes)	0.10 (0.04–0.24)	0.04	**0.001 ***
Availability of laboratory coats or gowns clinicians and support staff (Yes)	0.33 (0.10–1.07)	0.19	0.06
Personal protective garments were retained in the facility (Yes)	5.84 (3.09–11.02)	1.89	**<0.001 ^#^**
Clinician laboratory coats or gowns were washed within the facility (Yes)	2.97 (7.60–11.64)	2.07	0.12
Patient sheets and gowns were washed daily or when soiled (Yes)	0.10 (0.04–0.24)	0.04	**0.001 ***
Dirty linen or gowns transported in a designated container (Yes)	72.11 (20.22–257.14)	46.11	**<0.001 ^#^**
Availability of containers for needle disposal (Yes)	74.11 (19.46–282.27)	50.6	**<0.001 ^#^**
Biosafety waste collected in designated bins (Yes)	74.1 (23.70–231.70)	43.1	**<0.001 ^#^**
Biosafety waste removed at least daily from the area (Yes)	71.80 (20.75–248.45)	45.5	**<0.001 ^#^**
Availability of quarantine or isolation rooms in the department (Yes)	6.37 (1.61–25.16)	4.46	**0.008 ^#^**
Beds greater than 1 m apart (Yes)	0.99 (0.35–2.79)	0.52	1
The department was well ventilated (Yes)	2.70 (0.50–14.33)	2.3	0.24
Movement of people limited or restricted into the department (Yes)	0.23 (0.04–1.20)	0.19	0.08

**Bold** indicates significance at *p* ≤ 0.05. ^#^ denotes hospital infection control practices or environmental conditions that significantly increased bacterial loads while * denotes those that significantly decrease bacterial loads.

**Table 4 ijerph-18-06810-t004:** Multivariate analysis of the influence of hospital IPC practices and environmental conditions on bacterial loads in the study hospitals.

Hospital Infection Control Practices and Environmental Conditions	IRR (95% C.I.)	Robust Standard Error	*p*-Value
Mops and cleaning cloths stored wet (Yes)	4.48 (1.84–10.88)	2.03	**0.001 ^#^**
Frequency of cleaning and/or decontaminating the floor per day			
Twice	1.30 (0.95–1.80)	0.21	0.10
Thrice	0.40 (0.17–0.90)	0.17	**0.02 ***
Frequency of cleaning and/or decontaminating high-touch surfaces per day			
Thrice	1.53 (0.97–2.42)	0.36	0.06
Five times	0.64 (0.46–0.89)	0.11	**0.008 ***
Availability of hand wash station in the department (Yes)	0.88 (0.64–1.22)	0.15	0.47
Availability of soap at the hand wash station (Yes)	0.66 (0.50–0.86)	0.09	**0.003 ***
Are clinician laboratory coats or gowns washed within the facility? (Yes)	1.08 (0.69–1.69)	0.24	0.72
Personal protective garments were retained in the facility (Yes)	0.63 (0.23–1.72)	0.32	0.37
Biosafety waste collected in designated bins (Yes)	0.87 (0.46–1.67)	0.29	0.69
Biosafety waste removed at least daily from the department (Yes)	1.04 (0.61–1.76)	0.27	0.87
Movement of people is limited/restricted into the department (Yes)	0.98 (0.69–1.37)	0.16	0.90

**Bold** indicates significance at *p*-value ≤ 0.05. ^#^ denotes hospital infection control practices that significantly increased bacterial loads while * denotes those that significantly decreased bacterial loads.

## Data Availability

The data presented in this study are available on request from the corresponding author.

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
