# Peer review of "Ten Thousand-Fold Higher than Acceptable Bacterial Loads Detected in Kenyan Hospital Environments: Targeted Approaches to Reduce Contamination Levels"

_ijerph, 2021, doi:10.3390/ijerph18136810_

Round 1
Reviewer 1 Report
Odoyo et al. produced an interesting article on a study conducted in 2018 in five hospitals located in Kenya, showing the results coming from the nosocomial sampling of high-touch areas. The manuscript is well structured, according to the scientific method.
In order to improve this MS, please consider the following suggestions:
L (?) in introduction: “Healthcare-associated infections (HAIs) lead to prolonged hospital stays, increased
33 mortality”: is it possible to give a more up-to-date citation than in 1978?
L 47-48: Citation numbers have not been correctly indicated
L 85: ; -> :
L 328: Citation number has not been correctly indicated
It is regrettable to note that very often the bibliography is not up-to-date (dated citations) and that this important MS consists of only 26 bibliographical notes. It would be an excellent opportunity to enrich the bibliography, given the importance of the topic of the article, which is very well developed.
I would like to ask you to include, in the discussion/conclusions section, a detailed examination of the ways in which the adoption of the 2010 guidelines impacts on the pre-existing situation in your Country and the potential implications of Public Health expected or already reached.
No self-citations have been detected, OK.
Thank you for your efforts in perfecting the article.
Author Response
We want to thank the reviewer 1 for his or her comments and suggestions, which we found to be very helpful in improving the quality of the manuscript. Attached are the point-by-point responses to the comments raised. Changes made in the manuscript have been highlighted in red and tracked, and responses in this cover letter are in bold.

Reviewer 2 Report
This study tried to determine the levels and variations in 16 bacterial contamination of high-touch surfaces in five Kenyan hospitals and identify the contributing modifiable risk factors. The study also advised the study hospitals to reduce the bacterial loads by taking some effective measures. This paper made a quantitative analysis of high-touch surfaces’ bacterial contamination to ensure the accuracy of the results, which is innovative. Suggestions are provided as follows:
1.In "1. Introduction", the background of this research is introduced, I suggest the author to add a brief note about the background of research approach.
2.In "2. Materials and Methods", the materials and methods were introduced. I suggest the author to add a research frame, which can express the study process explicitly.
3.In "2.2 Environmental sampling", I suggest the author to make a list about "high-touch areas" which can show the areas that patients and clinical staff frequently touched clearly.
4.In "Table 1: Characteristics of IPC practices and bacterial loads", by taking hospital infection control practices, most values of " Total No. of items/equipment sampled" reduced, but some values still raised. I suggest the author to give an explanation about this phenomenon.
5.In "table 3", "Biosafety waste collected in designated bins (Yes) " appeared twice, but their IRR and Standard error is different. Did the meaning of those two items have any difference? If it did, please explain the difference about them.
6.There are some details should be corrected, such as some punctuation and unnecessary spaces.
In any case, this paper's starting point is so meaningful that this article can provide a certain reference for infection prevention and control. This research also has strong logic and clear structure. In the future, author can explore the types and distributions of the bacteria detected in high-touch surfaces and identify the contributing modifiable risk factors. I hope the suggestions can make the paper more scientific and rigorous.
Author Response
We want to thank the reviewer 2 for his or her comments and suggestions, which we found to be very helpful in improving the quality of the manuscript. Attached are the point-by-point responses to the comments raised. Changes made in the manuscript have been highlighted in red and tracked, and responses in this cover letter are in bold.

Round 2
Reviewer 1 Report
I recognise the efforts to make this article even more appreciable by the reader interested in this topic, enriching and updating, among other things, the bibliography. Discussions are now also related to behavioral criticalities of health professionals and to the impossibility of providing precise data on the situation that existed before the entry into force of the guidelines, as assessment and systematic monitoring were not carried out. All my suggestions have been accepted as part of a MS that is properly structured and characterized by precision and rationality.
Thank you.